# Research on Fault Diagnosis of Rolling Bearings Based on Variational Mode Decomposition Improved by the Niche Genetic Algorithm

**DOI:** 10.3390/e24060825

**Published:** 2022-06-14

**Authors:** Ruimin Shi, Bukang Wang, Zongyan Wang, Jiquan Liu, Xinyu Feng, Lei Dong

**Affiliations:** 1School of Mechanical Engineering, North University of China, Taiyuan 030051, China; 20150019@nuc.edu.cn (R.S.); 19840219@nuc.edu.cn (Z.W.); fengxinyu2017@nuc.edu.cn (X.F.); 2Department of Science and Technology Development, Taiyuan Institute of China Coal Technology Engineering Group, Taiyuan 030006, China; bkwang@126.com (B.W.); mkyljq@126.com (J.L.)

**Keywords:** variational mode decomposition, the Niche Genetic Algorithm, rolling bearing, Shannon entropy, fault diagnosis

## Abstract

Due to the influence of signal-to-noise ratio in the early failure stage of rolling bearings in rotating machinery, it is difficult to effectively extract feature information. Variational Mode Decomposition (VMD) has been widely used to decompose vibration signals which can reflect more fault omens. In order to improve the efficiency and accuracy, a method to optimize VMD by using the Niche Genetic Algorithm (NGA) is proposed in this paper. In this method, the optimal Shannon entropy of modal components in a VMD algorithm is taken as the optimization objective, by using the NGA to constantly update and optimize the combination of influencing parameters composed of α and K so as to minimize the local minimum entropy. According to the obtained optimization results, the optimal input parameters of the VMD algorithm were set. The method mentioned is applied to the fault extraction of a simulated signal and a measured signal of a rolling bearing. The decomposition process of the rolling-bearing fault signal was transferred to the variational frame by the NGA-VMD algorithm, and several eigenmode function components were obtained. The energy feature extracted from the modal component containing the main fault information was used as the input vector of a particle swarm optimized support vector machine (PSO-SVM) and used to identify the fault type of the rolling bearing. The analysis results of the simulation signal and measured signal show that: the NGA-VMD algorithm can decompose the vibration signal of a rolling bearing accurately and has a better robust performance and correct recognition rate than the VMD algorithm. It can highlight the local characteristics of the original sample data and reduce the interference of the parameters selected artificially in the VMD algorithm on the processing results, improving the fault-diagnosis efficiency of rolling bearings.

## 1. Introduction

As an important component of rotating machinery, rolling bearings play a vital role in the normal operation of the machine. Once a failure occurs, it will have a huge impact on production safety. Therefore, it is necessary to collect and accurately identify faults in the early stages of failure. Furthermore, it is necessary to replace or repair damaged bearings to avoid cascading failures [1,2,3,4].

During the acquisition process, the vibration signal will be affected by many factors such as load, friction, and shock. It will then show non-stationary and non-linear characteristics. In response to the shortcomings of traditional signal analysis methods, Konstantin Dragomiretskiy et al. proposed a self-adaptive, quasi-orthogonal, and completely non-recursive decomposition method—the Variational Mode Decomposition (VMD) method—in 2014 [5]. This method transforms signal decomposition into a constrained variational problem, adaptively decomposing the signal into the sum of several Intrinsic Mode Function (IMF) components. It also overcomes problems in Empirical Mode Decomposition (EMD) [6] and traditional signal analysis methods, which have the problems of modal aliasing, inaccurate components and similar frequency components. The Variational Mode Decomposition (VMD) method has the advantages of high computing efficiency, modal stability, and robustness. Based on this, it was applied to the fault diagnosis of mechanical equipment [7,8,9,10].

It is known that the decomposed result of the VMD algorithm is affected by the selection of parameters, such as the secondary penalty factor α (balance constraint parameter) and the number of modal components K, when processing the signal, which makes the algorithm largely affected by human experience [11]. Yi et al. [12] applied the particle swarm optimization (PSO) algorithm into the parameter selection of VMD, with the cross-correlation coefficient between the decomposed mode component and the original signal being regarded as an evaluation index. Lian et al. [13] used a series of indicators, such as permutation entropy, to judge the decomposition results, then constantly adjusted the mode number until the appropriate K value is obtained. However, running VMD repeatedly increased the cumulative error and decreased efficiency. Li et al. [14] chose the initial mode number and the most suitable number by peak searching and the similarity principle, then combined the similar modes to enhance the fault feature, although the initial mode number was still chosen artificially. Zhang et al. [15] proposed a parameter-adaptive VMD method based on a grasshopper optimization algorithm (GOA) which constructed a weighted kurtosis index as an optimized objective to select the VMD parameters. The method took the mode frequency bandwidth into account, but it ignored the effect of the penalty factor. In [16], a coarse-to-fine decomposing strategy was applied into the VMD. The balance parameter and the number of the decomposed modes were used to evaluate the selection of the parameter. Ni et al. [17] established two nested statistical models, namely the generalized Gaussian cyclostationary model and generalized Gaussian stationary model, to characterize the fault vibrations, then calculated the statistical indicator and threshold as the criteria for parameter optimization. Liang et al. [18] selected the envelop entropy and Renyi entropy as fitness functions, using the multi-island genetic algorithm (MIGA) algorithm to search the most suitable VMD parameters K and α. He et al. [19] applied an artificial bee colony algorithm (ABC) to find the optimal parameters of VMD. A novel index called syncretic impact index was used to determine the hyper-parameters. Li et al. [20] constructed a new objective function, the maximum average envelope kurtosis, to reduce the impact of random noise and determine the mode number and quadratic penalty term adaptively through an intelligent optimization algorithm. Wang et al. [21] applied cross-correlation theory into the determination of the penalty factor, but ignored the influence of the penalty factor.

Based on the above literature, this paper proposes a rolling-bearing fault-diagnosis method based on Variational Mode Decomposition improved by the Niche Genetic Algorithm (NGA-VMD). In this method, the optimal Shannon entropy of modal components in the VMD algorithm is taken as the optimization objective, by using the NGA to constantly update and optimize the combination of influencing parameters composed of α and K so as to minimize the local minimum entropy. The energy feature conducted by the entropy factor is then extracted from the original fault signal. Finally, the feature factor is fed into a particle swarm optimized support vector machine (PSO-SVM) [22] classification model to identify the different fault patterns in the rolling bearing. The main work and contributions of this paper can be summarized as follows:(1)The NGA is introduced into the VMD to optimize the selection of the mode number K and penalty factor α.(2)Compared to VMD and EMD, the effectiveness and accuracy of the NGA-VMD is verified.(3)The NGA-VMD and PSO-SVM are combined into an effective fault-diagnosis method.

The rest of the paper is organized as follows. In Section 2, the idea of NGA-VMD is introduced. The simulation signal analysis is described in Section 3. In Section 4, the diagnostic process based on the optimized algorithm is introduced. The measured signal in the rolling-bearing experiment system is analyzed in Section 5. The final conclusions are given in Section 6.

## 2. Theoretical Basis

### 2.1. Variational Modal Decomposition

VMD is a non-recursive algorithm that solves and constructs variational problems as the main overall framework. In the process of iteratively solving a variational model, the algorithm uses the alternating multiplier method to continuously update the IMF and its center frequency. IMF demodulates to the corresponding base frequency band and, finally, extracts each IMF and its corresponding center frequency, effectively implementing adaptive decomposition of the original input signal.

The IMF is defined in the VMD algorithm as an AM-FM signal, and its modal function can be regarded as a harmonic signal. The specific expression is as follows:(1)$$u_{k}\left(t\right)=A_{k}\left(t\right)\cdot \cos\left[\phi_{k}\left(t\right)\right]\;$$
where $A_{k}\left(t\right)$ is the instantaneous amplitude and $u_{k}\left(t\right)$ is the instantaneous frequency, $\omega_{k}\left(t\right)=\phi_{k}\left(t\right)$;

The expression of the optimization variational model is as follows:(2)$$\{\begin{array}{l} min\{\sum_{k}∥\partial_{t}\left[\left(\delta \left(t\right)+\frac{j}{\pi t}\right)u_{k}\left(t\right)\right]e^{-j\omega_{k}\left(t\right)}{∥}_{2}^{2}\}\\ \sum_{k}u_{k}=f \end{array},\;$$
where $\left\{u_{k}\right\}=\left\{u_{1},u_{2},\cdots ,u_{K}\right\}$ is the decomposed *K* modal components and $\left\{\omega_{k}\right\}=\left\{\omega_{1},\omega_{2},\cdots ,\omega_{K}\right\}$ is the corresponding center frequency of $\left\{u_{k}\right\}$.

By introducing the Lagrange multiplication operators λ(t), an extended Lagrange expression is constructed, and the constrained problem is transformed into a non-constrained sub-optimization problem:(3)$$\Gamma \left(\left\{u_{k}\right\},\;\left\{\omega_{k}\right\},\;\lambda \right)=\alpha \underset{k}{\sum }∥\partial_{t}\left[\left(\delta \left(t\right)+\frac{j}{\pi t}\right)u_{k}\left(t\right)\right]e^{-j\omega_{k}t}{∥}_{2}^{2}+\;\;\;\;∥f\left(t\right)-\sum_{k}u_{k}{\left(t\right)∥}_{2}^{2}+\langle \lambda \left(t\right),f\left(t\right)-\sum_{k}u_{k}\left(t\right)\rangle ,$$

Based on the above augmented Lagrange expression, the saddle point of the multiplication operator alternating direction method is used to obtain its saddle point, and iteratively updates $\left\{u_{k}\right\}$ and $\left\{\omega_{k}\right\}$ continuously to obtain the optimal solution of the variational model. Specific conversion solutions were presented in the
literature [5].

### 2.2. NGA-VMD Algorithm

The decomposition result of VMD is largely affected by the value α and the preset K value. Although the introduced value α can ensure the reconstruction accuracy of the signal, as the value increases, the bandwidth of the modal component will also decrease; the K value needs to be set manually, and there is great uncertainty. If it is too large, then it will lead to over-decomposition, false modes appear, and too many decomposed signals are irregular; if it is too small, it will lead to loss of signal components and modal aliasing.

As an improved search optimization algorithm, the Niche Genetic Algorithm (NGA) allows individuals to evolve in a specific living environment, so that they can find all the optimal solutions to the problem [23]. Based on the NGA good global search ability and high convergence speed, the NGA is introduced to VMD to optimize the selection of the optimal parameter combination of the α and K values. At the same time, genetic operators are used before selection as an elite retention strategy to ensure the convergence of the NGA, retain the best genes to the greatest extent, and avoid the loss of elite individuals [24].

It is also necessary to define a fitness function when using the NGA to search for the influence parameters of the VMD algorithm. In terms of evaluating the sparse characteristics of signals, the Shannon entropy has obvious advantages as an evaluation standard [25]: the higher the degree of uncertainty of a signal, the greater its entropy value. The original signal is demodulated to obtain the envelope signal probability distribution sequence. After calculation, the entropy value of $e_{j}$ is the envelope entropy, which can be used to represent the sparse characteristics of the original input signal. This paper uses the signal envelope entropy to evaluate the sparse characteristics of the signal.

Zero mean signal x(j),j=1,2,⋯,N’s envelope entropy is *E*, which can be presented as follows:
(4)$$\left\{\begin{array}{c} E_{e}=-\sum_{j=1}^{N}{\mathrm{lge}}_{j}\\ e_{j}=\frac{a(j)}{\sum_{j=1}^{N}a\left(e_{j}\right)} \end{array},\right.$$
where a(j)  is the zero-mean signal and the envelope signal x(j) obtained after Hilbert transform; and $e_{j}$ is the normalized form of a(j).

In each cycle of NGA optimization, all the entropy values of the envelope after VMD processing as $u_{k}$ are calculated. Define the one with the smallest entropy value as the local minimum entropy value $E_{e\min}$, which will be used as the fitness value in the optimization. During the optimization process, the global optimal solution in evolutionary generation is found through continuous updates, and the corresponding (K, α) and combinations ($u_{k}$) are extracted.

## 3. Simulation Signal Analysis

In order to verify the superiority of the NGA-VMD algorithm without loss of generality, a multi-component simulation experimental signal was constructed for analysis. The specific expression of the simulation signal is:(5)$$\left\{\begin{array}{c} X\left(t\right)=x_{1}\left(t\right)+x_{2}\left(t\right)+x_{3}\left(t\right)+x_{4}\left(t\right)+s\left(t\right)\\ x_{1}\left(t\right)=2\;cos\left(2\pi f_{1}t\right)\\ x_{2}\left(t\right)=4\;cos\left(2\pi f_{2}t\right)\\ x_{3}\left(t\right)=8\;cos\left(2\pi f_{3}t\right)\\ x_{4}\left(t\right)=16\;cos\left(2\pi f_{4}t\right)\\ s(t)=0.3rand(1,n),n=length(t) \end{array}\right.$$
where the frequencies $f_{1}\sim f_{4}\;$ corresponding to the four component signal components are 2 Hz, 24 Hz, 120 Hz, and 288 Hz; Gaussian white noise with a variance of 0.3 s(t) were added as the superimposed interference signal.

The time-domain waveforms of the original simulation signal are shown in Figure 1a, and its component signals are shown in Figure 1b:

No features can be seen from Figure 1a. The NGA-VMD algorithm was used to decompose the simulation signals. Figure 2 is a graph of the minimum entropy of simulated signals under different evolutional generations during the optimization of the VMD input parameters by the NGA. The local minimum entropy value of the modal component 0.217 appeared in the 6th generation. The optimal input parameter combination (K, α) = (4, 645) was searched. Therefore, the K value in the VMD was set to 4 and the second penalty factor α set to 645 to perform modal decomposition on the simulation signal. Figure 3 is a component time-domain waveform diagram obtained by analyzing and processing the simulated signal through the NGA-VMD algorithm.

By executing the NGA-VMD algorithm, four IMF components were obtained. Comparing Figure 3 and Figure 4, it can be found that the waveforms of the IMF component and the original component signal have good consistency. Compared with traditional VMD, this algorithm achieves fast adaptive decomposition of signals, avoids the occurrence of over-decomposition and under-decomposition, and greatly reduces the interference of human factors.

## 4. Diagnostic Process

In order to realize the intelligent diagnosis of rolling-bearing faults, increase the accuracy and speed of recognition, and reduce the impact of human factors on the diagnosis results, the energy features extracted from the original fault signal are calculated as the input vector of the PSO-SVM, and a “One-on-One” classifier structure is adopted. The diagnosis process is shown in Figure 5. The specific steps are as follows:(1)Collect and load the operating data of each state of the rolling bearing;(2)Use the NGA-VMD algorithm to optimize the collected rolling-bearing experimental data to obtain the optimal combination of influencing parameters, and realize the collected signals by decomposition to obtain *K* modal components, where is k _=_ 1, 2,⋯, K(3)Calculate the entropy value containing *u*_*k*_, to construct the corresponding energy eigenvector T; construct the value of T as follows:
(6)$$\{\begin{array}{c} T=\left[\frac{E_{1}}{E},\frac{E_{1}}{E},\cdots ,\frac{E_{k}}{E}\right]\\ E=\sqrt{\sum_{k=1}^{K}{\left|E_{k}\right|}^{2}} \end{array},k=1,2,\cdots ,K,\;$$

(4)Input the obtained T value into the PSO-SVM as an input vector and complete the fault type identification and classification of the rolling bearing through the PSO-SVM.

## 5. Application Case Analysis

In order to further illustrate the effectiveness of the NGA-VMD algorithm, experiments were performed using bearing data from the Bearing Data Center of Case Engineering, Electrical Engineering Laboratory, Case Western Reserve University.

The test bench is shown in Figure 6. The test bearings were installed at both ends of the motor. The drive end bearings were 6205-2RS SKF deep-groove ball bearings. Single-point faults were processed by electric spark machining technology. The inner and outer rings of the test bearings and rolling elements were introduced. The vibration data were collected by an acceleration sensor placed in the direction of the radial load of the test bearing. The experiment took four states to verify: normal bearing (NOR), inner ring fault (IRF), outer ring fault (ORF) and rolling element fault (REF). In order to improve the accuracy of the data, the real industrial production environment was highly simulated without loss of generality, and the collected signal was mixed with a Gaussian white noise component with a signal-to-noise ratio of −1 dB The specific data of bearings and experimental conditions are shown in Table 1 and Table 2.

Figure 7 shows the collected bearing data waveform and spectrum of different fault modes. Analysis shows that the fault characteristics of rolling bearings cannot be distinguished in the collected signals in different states, and there are obvious peaks near different frequency bands of the spectrum of each signal. In addition, the influence of noise cannot be used to distinguish fault features. Therefore, the signals need to be decomposed by corresponding algorithms.

Figure 8 shows the NGA-VMD decomposition results of bearing inner ring fault signals. In order to improve the accuracy of the algorithm and fault classification, the number of signal sampling points was properly screened before signal processing. Table 3 displays the calculated average entropy of the signal at different sampling points. Analysis shows that as the signal length increases, the entropy value of the signal will also gradually decrease. After the number of sampling points exceeds 2048, the entropy value will be stable, and the entropy value will balance after 4086. Therefore, the data length was taken as 4086 in the subsequent PSO-SVM training and testing.

For the vibration signals collected in the four states of the bearing: NOR, IRF, RRF and BEF, 40 sets of data were taken in each of the four states, for a total of 160 sets of experimental data, of which the data sample length is 4086. For bearing signal data collected in different states, 10 groups were randomly selected from the samples in each state, a total of 40 groups were used as training samples for PSO-SVM, and the remaining 120 groups of data were used as test samples.

NGA-VMD decomposition was performed on the randomly selected training samples. Figure 9 shows the NGA-VMD algorithm optimizing the optimal parameters of the collected bearing signals in four different states. The results for searching optimizations are shown in Table 4.

NGA-VMD decomposition was performed on the training sample data. Each training sample obtained K modal components and calculated the modal entropy values. The entropy values obtained from each training sample were combined to form the corresponding T and then the normalization process was used as an input to the PSO-SVM for training. Due to space limitations, only some sample data of T value after processing calculation are listed in Table 5.

The fault classification and identification results for rolling bearings under different signal processing modes are listed in Table 6. Figure 10 shows the trend of fault recognition rate of the four algorithms. The analysis shows that no matter which signal processing method is used, the bearing can achieve 100% recognition under normal conditions; however, when the bearing is in a fault state, the difference in the accuracy rate of fault recognition is obvious. The NGA-VMD algorithm is used for modal decomposition. The average correct rate of fault recognition is 99.17%. The result is 1.67% better than that of the other optimization algorithm (GOA-VMD). The traditional VMD algorithm or EMD algorithm were used for signal processing. The average correct rate of fault recognition was significantly reduced. The average correct rates of fault recognition were 94.17% and 87.5%, respectively.

The comparison results of the three different signal processing methods show that the NGA-VMD algorithm proposed in this paper performs signal modal decomposition, which can more easily and effectively extract fault features, and its fault recognition rate is improved. According to experimental analysis and testing, the NGA-VMD algorithm achieves better than the optimization of traditional algorithms, when processing the same experimental data. Moreover, its algorithm processing efficiency and PSO-SVM classification time are significantly better than other similar algorithms.

In order to study the impact of different types of training samples on the classification and recognition results of the running status of rolling bearings, the experimental data in the four states of NOR, TRF, ORF, and BEF were used to randomly select different proportions of data as training samples to complete the training of the PSO-SVM. The remaining data were tested to complete fault classification and identification. Figure 11 shows the fault recognition rate under different training samples. The analysis shows that when the training sample is less than 5%, the NGA-VMD decomposition proposed in this article is used to classify the rolling bearing faults through PSO-SVM, and the classification accuracy rate is less than 80%. When the sample proportion exceeds 10%, the classification accuracy rate is fast. The improvement is stable at 25%; when the sample proportion exceeds 40%, the classification accuracy rate can reach 100%. Increasing the proportion of training samples can make the established PSO-SVM prediction model more accurate, and the correct rate of fault recognition can be improved accordingly.

## 6. Conclusions

(1)This paper proposes the NGA-VMD algorithm to reduce the influence of the two key parameters (α, K) of the VMD algorithm. The two affected parameters are optimized for the VMD algorithm to implement signal processing more effectively and accurately. The NGA-VMD algorithm, as a new signal processing method, greatly reduces the interference of human factors on the processing results, has better noise robustness and data processing efficiency, and can better highlight the local characteristics of the original sample data.(2)Simulation and analysis of experimental results show that relative to VMD and EMD, the NGA-VMD algorithm can achieve rapid adaptive signal decomposition, avoid the occurrence of over or under decomposition, and greatly reduce the interference of human factors. Under the same experimental conditions, the NGA-VMD algorithm performs modal decomposition, and the average correct recognition rate of faults is 99.17%, with the average correct recognition rates of GOA-VMD, VMD, and EMD algorithms being 97.5%, 94.17%, and 87.5%, respectively. The NGA-VMD algorithm takes 95.8 s, which is 7.62% faster than GOA-VMD, 49.9% faster than VMD, and 79.5% faster than EMD.(3)The NGA introduced in this paper realized the optimization of the VMD algorithm, combined with PSO-SVM to accurately complete the fault identification and classification of rolling bearings, and obtained a good diagnostic effect. It provides a more practical solution for the analysis and treatment of other types of mechanical faults and is worth further in-depth research.

## Figures and Tables

**Figure 1 entropy-24-00825-f001:**
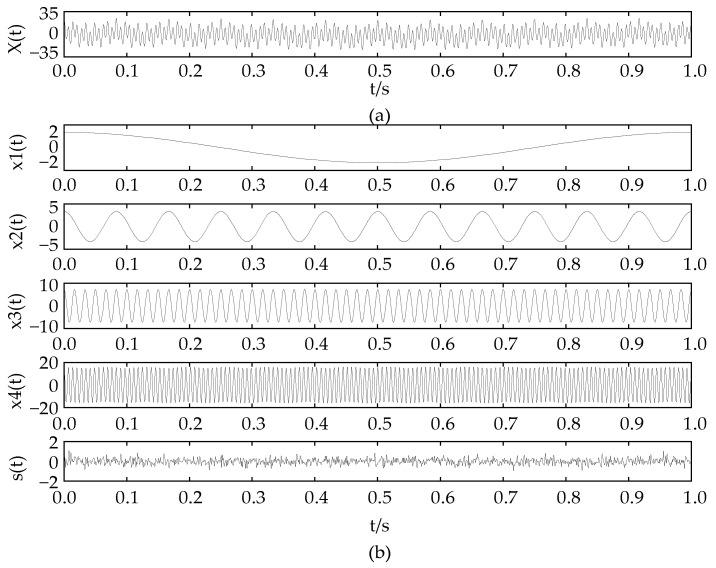
Time-domain waveform of the original simulation signal (**a**) and its component signals (**b**).

**Figure 2 entropy-24-00825-f002:**
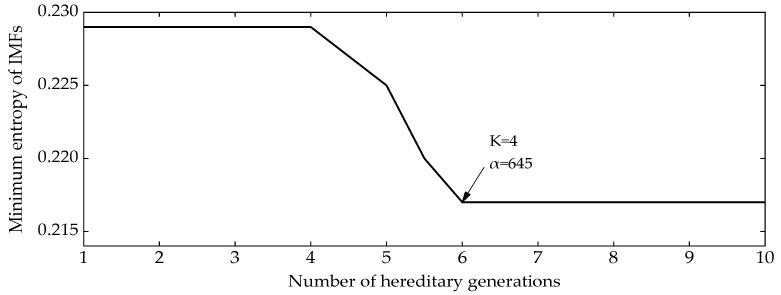
Minimum entropy diagram of different number of evolutional generations in optimization process.

**Figure 3 entropy-24-00825-f003:**
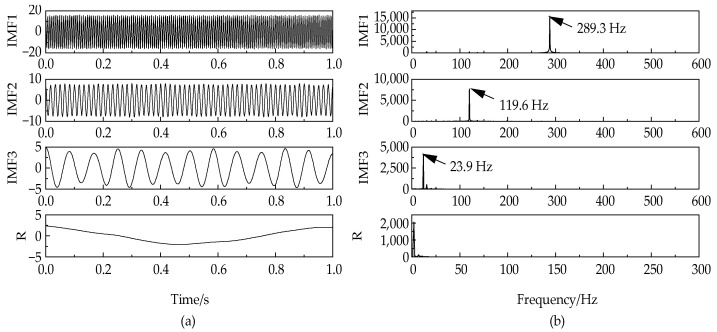
Time-domain diagram (**a**) and frequency-domain diagram (**b**) of VMD.

**Figure 4 entropy-24-00825-f004:**
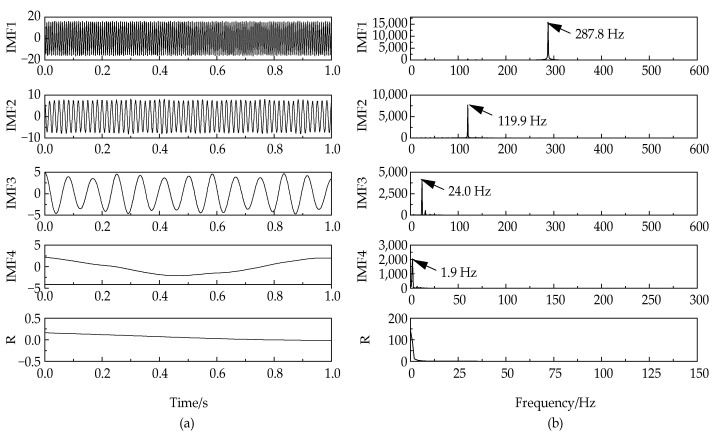
Time-domain diagram (**a**) and frequency-domain diagram (**b**) of NGA-VMD.

**Figure 5 entropy-24-00825-f005:**
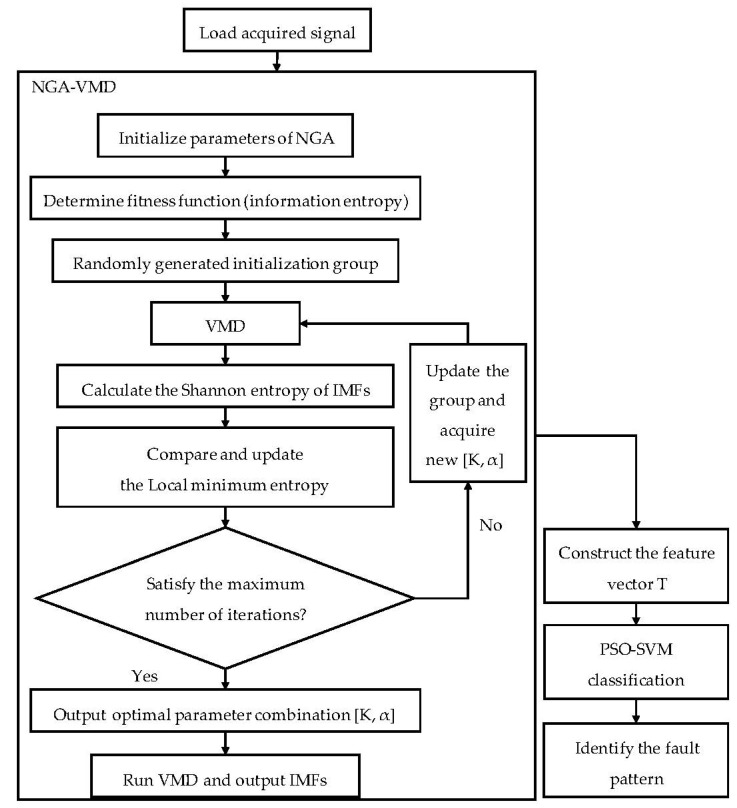
Flow chart of diagnosis.

**Figure 6 entropy-24-00825-f006:**
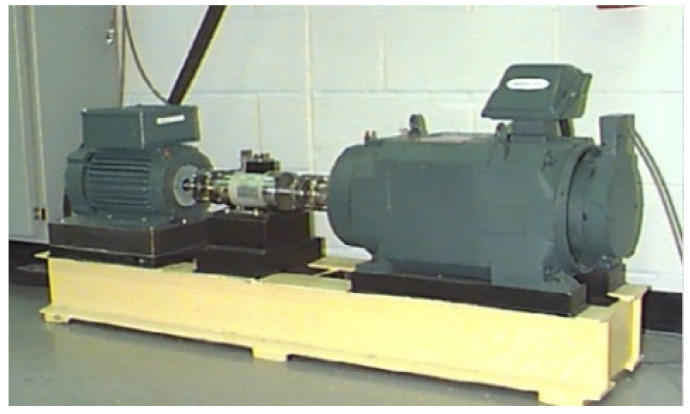
The test bench for the bearing fault experiment.

**Figure 7 entropy-24-00825-f007:**
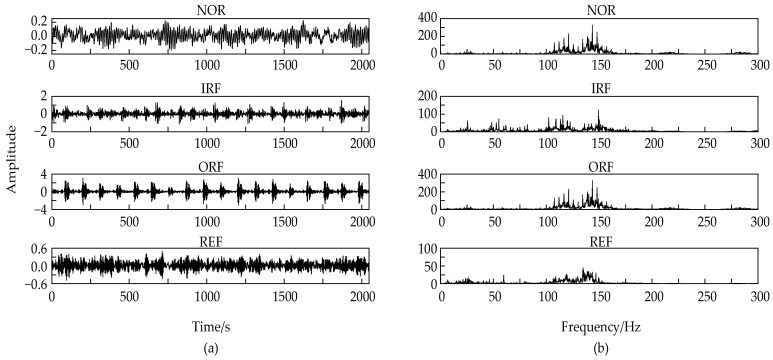
Time domain diagram (**a**) and spectrum diagram (**b**) of different bearing states.

**Figure 8 entropy-24-00825-f008:**
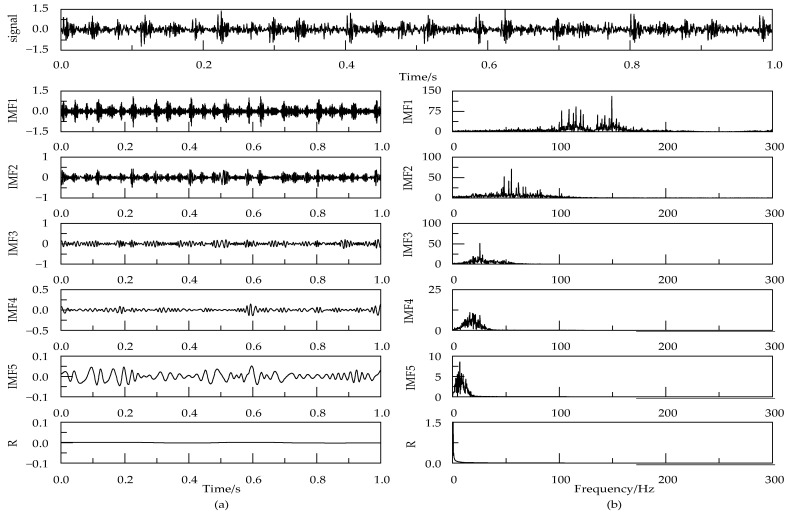
Time-domain diagram (**a**) and spectrum diagram (**b**) of IMFs component obtained using NGA-VMD for inner ring fault signal.

**Figure 9 entropy-24-00825-f009:**
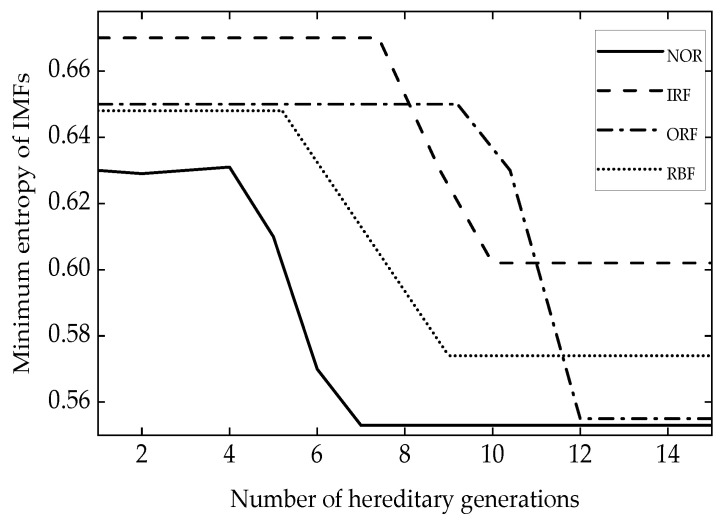
Parameter optimization under four kinds of bearing conditions.

**Figure 10 entropy-24-00825-f010:**
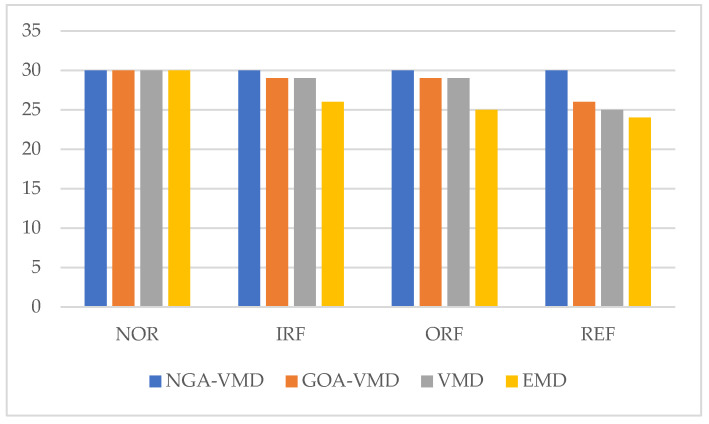
Comparison of fault recognition rate of four algorithms.

**Figure 11 entropy-24-00825-f011:**
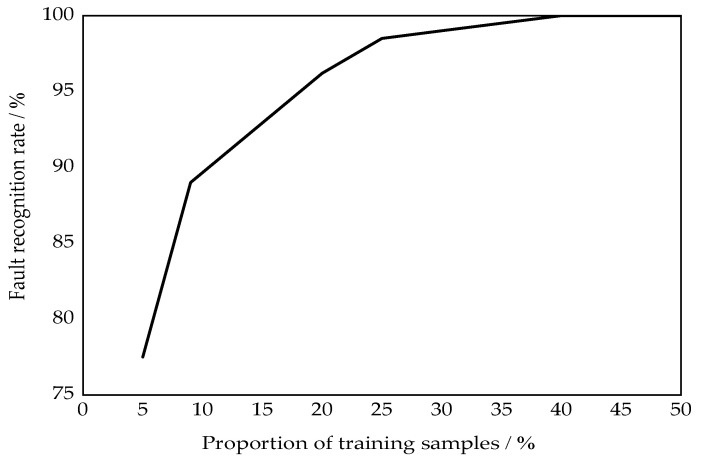
Fault recognition rate under different proportions of training samples.

**Table 1 entropy-24-00825-t001:** Parameters of the testing bearing.

Middle Diameter of Bearing	Diameter of the Roller	Contact Angle	Number of Rollers
38.5 mm	8 mm	0°	9

**Table 2 entropy-24-00825-t002:** Parameters of the experiment.

Rotational Speed	Diameter of Fault Point	Sampling Frequency	Initial Number of Sampling Point
1797 r/min	0.1778 mm	12 kHz	2048

**Table 3 entropy-24-00825-t003:** Average entropy of signal at different sampling points.

Sampling Points	Average Entropy
512	0.863
1024	0.851
2048	0.644
4086	0.631
8192	0.620

**Table 4 entropy-24-00825-t004:** Results for searching optimization.

State	Local Minimum Entropy	(K, α)
NOR	0.5602	(4, 860)
IRF	0.5998	(7, 1000)
ORF	0.5473	(9, 1200)
REF	0.5728	(5, 600)

**Table 5 entropy-24-00825-t005:** T values of partial bearings in four states.

State	Sample	T								
		*E* _1_	*E* _2_	*E* _3_	*E* _4_	*E* _5_	*E* _6_	*E* _7_	*E* _8_	*E* _9_
NOR	1	0.2159	0.3151	0.2621	0.2319	—	—	—	—	—
	2	0.2239	0.3381	0.2113	0.2245	—	—	—	—	—
IRF	1	0.1023	0.1802	0.2634	0.3011	0.3689	0.3731	0.3623	—	—
	2	0.1076	0.1864	0.2788	0.2193	0.3514	0.3677	0.3799	—	—
ORF	1	0.1143	0.2001	0.3114	0.2987	0.4567	0.4312	0.4501	0.3644	0.3127
	2	0.1533	0.2409	0.3002	0.3233	0.4763	0.4772	0.3986	0.3876	0.3321
REF	1	0.1556	0.2192	0.4018	0.4871	0.1984	—	—	—	—
	2	0.1848	0.2997	0.3851	0.4639	0.1869	—	—	—	—

**Table 6 entropy-24-00825-t006:** Fault classification and identification results for rolling bearings under different signal processing modes.

State			NOR	IRF	ORF	REF	Average Accuracy	Running Time/s
Number of samples			30	30	30	30	99.17%	95.8
Signal processing	NGA-VMD	NOR	30	0	0	0		
		IRF	0	30	0	1		
		ORF	0	0	30	0		
		REF	0	0	0	29		
		Classification accuracy	100%	100%	100%	96.67%		
	GOA-VMD	NOR	30	0	0	0	97.50%	103.1
		IRF	0	29	1	1		
		ORF	0	0	29	0		
		REF	0	1	0	29		
		Classification accuracy	100%	96.67%	96.67%	96.67%		
	VMD	NOR	30	0	0	0	94.17%	143.6
		IRF	0	29	0	0		
		ORF	0	1	28	4		
		REF	0	0	2	26		
		Classification accuracy	100%	96.67%	93.33%	86.67%		
	EMD	NOR	30	0	0	0	87.50%	171.9
		IRF	0	26	2	3		
		ORF	0	4	25	2		
		REF	0	0	3	24		
		Classification accuracy	100%	86.67%	83.33%	80.00%		

## Data Availability

The data presented in this study are available on request from the corresponding author.

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
