# Peer review of "Research on Fault Diagnosis of Rolling Bearings Based on Variational Mode Decomposition Improved by the Niche Genetic Algorithm"

_entropy, 2022, doi:10.3390/e24060825_

Round 1
Reviewer 1 Report
In this work, a new combination of methods (Niche Genetic Algorithm and Variational Mode Decomposition method) is presented for fault diagnosis of rolling bearings. Although the work is interesting, some aspects are not clear.
The contribution is not clear. Is your work only another solution for a solved problem? Please state clearly the advantages over other works already presented in the literature.
Lines 44-45: Check the sentence.
The application of the Niche Genetic Algorithm vs other optimization algorithms is not justified; in fact, a comparison in quantitative (or at least in a qualitative way) is required.
Please justify the values used in equation (5). What is the impact on the results if close frequencies are presented (e.g., 24 Hz and 25 Hz)? Please provide the results. Noise immunity should be also discussed; please add and discuss results for higher values of noise.
More details and results about the NGA application are necessary. Readers cannot reproduce your results with the information provided.
Line 176: It is not clear how you obtain K=4.
Lines 187-188: That information lacks of quantitative results and is subjective. Please provide numerical errors in both time and frequency domains (they are necessary).
Lines 188-191: That information lacks of evidence. Please provide the obtained results if the classical VMD method is used. Errors in time and frequency domains are necessary; also, discuss computational time.
A comparison with the results presented in the state of the art for the same database has to be included.
Partial results for the flowchart presented in Figure 4 have to be included in order to see the evolution of the proposed methodology.
For results presented in Tables 5 and 6, please add graphs in order to see maxima and minima values, trends, etc.
The theoretical basis for PSO-VSM is not provided.
It is not clear why the Authors claim that fault characteristics cannot be obtained from the results of Figure 7. Figure 7 does show differences in the spectra, which can be recognized by multiple pattern recognition algorithms.
The methods presented in Table 6 cannot be reproduced. Please add more information. Also, the results in Table 6 are unfair and worthless. Results from an optimized method cannot be compared with the ones obtained by non-optimized methods (it is evident which will be the best). Optimized methods have to be compared with optimized methods. Please discuss in detail this point and do a fair comparison.
Include the extracted IMFs for the signals presented in Figure 6. For these IMFs, include their spectrum in order to correlate the frequency components presented in Figure 7.
Author Response
Dear Reviewers:
Thank you for your letter and for the reviewers’ comments concerning our manuscript 1757794. Please see the attachment for the main corrections in the paper and the responds.
Sincerely yours,
Lei Dong

Reviewer 2 Report
Dear authors,
I read your article with interest. It seems convincing.
Nevertheless, there are some improvements to be made.
The first one is typographical. There is no capital letter after a comma, missing space after a dot.
Something seems stange about ORF and BFF [ outer ring fault (ORF) and rolling body fault (BFF) ]. Figures 6, 7, 8 ... mention ORF and BEF.
I think you should do more experiments with bearings with a non-zero contact angle. This would prove irrevocably that this diagnostic method is relevant.
Good luck for the next step.
Author Response
Dear Reviewer:
Thank you for your letter and for the reviewers’ comments concerning our manuscript 1757794. Please see the attachment for the main corrections in the paper and the responds.
Sincerely yours,
Lei Dong

Reviewer 3 Report
The topic of the work is relevant because the violation of the operating mode and the breakdown of complex technical systems is an important problem for the present. The use of diagnostic methods makes it possible to exclude options for the transition to an undesirable mode of operation and to reveal hidden defects. This can ultimately prevent catastrophic consequences when equipment breaks down.
1. Literary review is modern. 19 out of 25 literature sources are not older than 5 years.
The introduction provides an analysis of known results in the field of diagnostics of rolling friction units. The main provisions of the work are formulated.
2. The method of work is detailed and well described, although it is stretched over the entire text of the work, and not collected in one section, as is usually the case.
3. The results are also described in sufficient detail and are accompanied by high-quality illustrations and tabular data.
4. The positive aspect of the work is the verification of the selected diagnostic algorithm using a test bench (section 5).
5. The conclusions are based on the results obtained and are reliable and adequate.
Notes:
Figure 1. There is no description of positions a and b in the caption.
Table 6. Header and table itself on different pages. Must be placed on one.
Conclusion 2. Indicate how much % faster the proposed algorithm.
Author Response

(The authors gave the same response as above.)

Round 2
Reviewer 1 Report
Comments and suggestions have been properly addressed. This Reviewer recommends the manuscript acceptance.